# Non-polyalanine repeat mutation in *PHOX2B* is detected in autopsy cases of sudden unexpected infant death

**Atsushi Ueda**[1]☯, **Motoki Osawa**[1]☯*, **Haruaki Naito**[1], **Eriko Ochiai**[1,2], **Yu Kakimoto**[1]

**1** Department of Forensic Medicine, Tokai University School of Medicine, Isehara, Kanagawa, Japan,
**2** Department of Legal Medicine, Kitasato University School of Medicine, Sagamihara, Kanagawa, Japan

☯ These authors contributed equally to this work.

\* osawa@is.icc.u-tokai.ac.jp

## Abstract

### Background

Congenital central hypoventilation syndrome (CCHS), which is caused by *PHOX2B* with phenotypic variations, has a point of controversy: CCHS is putatively involved in autopsy cases of sudden unexpected infant death (SUID) including sudden infant death syndrome.

### Objective

The relation of CCHS to SUID cases was investigated by extensive genotyping of *PHOX2B*.

### Methods

We analyzed 93 DNA samples of less than one-year-old SUID cases that were autopsied in our department. Unrelated adult volunteers (*n* = 942) were used as the control.

### Results

No polyalanine tract expansion was detected in the SUID cases. The allelic frequencies of repeat contractions and SNP (rs28647582) in intron 2 were not significantly different from that in those control group. Further extensive sequencing revealed a non-polyalanine repeat mutation (NPARM) of c.905A>C in a sudden death case of a one-month-old male infant. This missense mutation (p.Asn302Thr), registered as rs779068107, was annotated to 'Affected status is unknown', but it might be associated with the sudden death.

### Conclusion

NPARM was more plausibly related to sudden unexpected death than expansions because of severe clinical complications. This finding indicates possible CCHS involvement in forensic autopsy cases without ante-mortem diagnosis.

**Data Availability Statement:** All relevant data are within the paper and its Supporting information files.

**Funding:** The authors received no specific funding for this work.

**Competing interests:** The authors have declared that no competing interests exist.

## Introduction

Congenital central hypoventilation syndrome (CCHS) is characterized by impaired ventilatory response to hypercapnia and hypoxemia during sleep, which usually depend on artificial ventilatory support. This autosomal dominant disorder is caused by the single gene of paired-like homeobox 2B (*PHOX2B*), which is in turn associated with various phenotypic disorders [1]. Most patients carry expansions of the repetitive alanine tract, ranging from 5 to 13 residues, which are usually generated by *de novo* mutation in meiosis. The normal allele is 20 times repeated, but a couple of contracted alleles are also prevalent [2]. Furthermore, the minor part carries non-polyalanine repeat mutation (NPARM) of frameshift, nonsense and missense, which more often result in severe complications of Hirschsprung disease and neural crest-derived tumors [3, 4].

The diagnosis of sudden infant death syndrome (SIDS) as a cause of death has decreased in mortality statistics for decades. One reason for the decrease is circumvention by examiners because of indistinguishable disorders such as accidental suffocation in bed. Other broad criteria such as sudden unexpected infant death (SUID) and sudden unexpected death in infancy are preferably chosen to indicate causes for such equivocal unclassifiable infant deaths [5, 6]. The involvement of CCHS in SIDS/SUID has been investigated for forensic autopsy cases because of its similarity of nocturnal onset.

No apparent involvement of CCHS in SIDS cases has been demonstrated in earlier studies [7, 8]. However, Rand et al. demonstrated significant allelic frequency difference of SNP (rs28647582) in intron 2 between SIDS and control groups of the Caucasian population [9]. Liebrechts-Akkerman et al. found significant association of repeat contractions with unclassified sudden infant death in a Dutch population [10]. Ventura et al. reported contraction in an infant autopsy case, but its causality was unclear [11]. Furthermore, a recent study has shown that SNP in the 3′ untranslated region affects the expression of PHOX2B molecules, which potentially impairs respiratory function via the neurological system [12]. In contrast, some negative reports also described that no pathogenic variation of *PHOX2B* was detected in cases of SIDS [13] or sudden unexpected death in epilepsy cases [14]. The association of CCHS remains controversial.

This report describes a specific investigation of the relation of *PHOX2B* variation with SIDS/SUID. The distribution of repeat alleles was examined in large groups of Japanese subjects: SUID subjects in the forensic department and healthy control individuals. Then, extensive sequencing of *PHOX2B* was performed for the SUID samples.

## Materials and methods

### Subjects

For this study, we analyzed 93 SUID samples from the Department of Forensic Medicine, Tokai University School of Medicine, during 17 years: 2005–2021. The criteria for selection of SUID were fundamentally as described previously [15]. These less than one-year-old 58 male and 35 female subjects had mean age of 3.4 months. Full autopsies were performed including examinations of histopathology, toxicology, clinical chemistry, and virus antibody titers [16]. As the cause of death, SIDS and suspected SIDS cases were 59, undetermined cases were 25, and infections of airways such as bronchitis were 9. Overlay and suffocation during co-sleeping were excluded as possible. As control subjects, unrelated adult volunteers (*n* = 942) were available [17].

This project was approved by the Ethical Committee of Tokai University Hospital. Written informed consent was obtained from the parents of SUID cases and the control subjects.

## Molecular analysis of *PHOX2B*

DNA was extracted from an oral swab of control subjects and blood at autopsy. An oligonucleotide primer set of 5′-aggtcccaatcccaaccccac-3′ and 5′-gaatccgggatggaggt gatg-3′ was designed to obtain the repetitive tract. Briefly, PCR amplification was performed using Tks Gflex DNA Polymerase (Takara Bio Inc., Kusatsu, Japan) from 10 ng of DNA as the template with 40 thermal cycles of 98˚C for 10 s, 60˚C for 15 s, and 68˚C for 10 s. The product was analyzed using a genetic analyzer (3500; Thermo Fisher Scientific, MA, USA).

The GC content of the repeat and its vicinity is very high, around 90%, which often inhibits amplification of sufficient amounts of the expanded alleles [2]. For an earlier study, we developed effective circumvention using bisulfite-treated DNA [18]. In addition to the PCR amplification described above, we applied this modified one to homozygous specimens.

Furthermore, for all SUID cases, the nucleotide sequence of three exons and two introns of *PHOX2B* was determined for PCR amplicons of 2999 bp using the primer set of 5′-gcgttgagctgtgcacatctc-3′ and 5′-gacgacaatagccttgggcct-3′. Direct sequencing of the products was carried using the Sanger method. The detected substitutions were surveyed on the web site of the National Center for Biotechnology Information Search database (https://www.ncbi.nlm.nih.gov/).

To determine allele frequencies of two substitutions detected in the SUID subjects, the amplified product-length polymorphism (APLP) analysis was employed to the control group [19], using the following primer sets; 5′- tcttcgctccaaagagccaa-3′, 5′-ttagtcttcgctccaaagaccgac-3′ and 5′- catactgctcttcactaaggcg-3′ for rs779068107, and 5′- agtcctggagcctcgggtta-3′, 5′- atatagtcctggagcctc gaaagg-3′ and 5′- tatttctgatcggccatggggc-3′ for rs28647582, where underline means replaced sites in the oligonucleotides.

## Next-generation sequencing (NGS) analysis

DNA libraries were constructed using the Ion AmpliSeq Library kit 2.0 (Thermo Fisher Scientific Inc.), according to the manufacturer's protocol. Briefly, Ion AmpliSeq Inherited Disease Panel (Thermo Fisher Scientific Inc.) was used as the primer set for all exon regions of 328 disease related genes. After amplification of emulsion PCR in the Ion OneTouch 2 instrument with the Ion PGM Template OT2 400 kit, sequencing was carried out using the Ion 316 Chip kit V2 BC (Thermo Fisher Scientific Inc.). The sequence data were analyzed using Torrent Suite Software ver. 5.2.2. The detected variants were subjected to the predictive *in silico* tool of Ion Reporter Software ver. 5.1.8 (https://ionreporter.thermofisher.com/ir/) including ClinVar for the causative genes of cardiovascular diseases such as QT prolongation and hypertrophic cardiomyopathy.

## Statistical analyses

Categorical variables of Hardy-Weinberg disequilibrium and allelic distribution were compared using either chi-squared tests or Fisher's exact test when appropriate. A *P* value of < .05 was inferred as statistically significant.

## Results

### Repetitive tract analysis

Table 1 summarizes the allele frequency of the polyalanine repeat in two groups. No expansion was detected in the control group (*n* = 942). Two types of 15-times and 13-times contractions

**Table 1. Frequency (number) of genotypes and alleles of the polyalanine repeat in the SUID and control groups.**

| | Genotype | | | | | Allele | | | |
|---|---|---|---|---|---|---|---|---|---|
| | 20/20 | 20/15 | 20/13 | 15/15 | Expansion | 20 | 15 | 13 | 15 +13 |
| SUID (n = 93) | 0.946 (88) | 0.032 (3) | 0.022 (2) | None | None | 0.973 (181) | 0.016 (3) *P* = .15 | 0.011 (2) *P* = .20 | 0.027 (5) *P* = .09 |
| Control (n = 942) | 0.920 (867) | 0.068 (64) | 0.007 (7) | 0.004 (4) | None | 0.958 (1805) | 0.038 (72) | 0.004 (7) | 0.042 (79) |

were detected with allele frequencies of 0.038 and 0.004, respectively, of which values were within the Hardy-Weinberg equilibrium ($\chi 2$ = 5.48, df = 2).

The high GC content of repetitive tract often inhibits amplification of expanded alleles. To optimize efficacy in raising potential hidden expansions, we applied an alternative analysis using bisulfite-treated DNA to all homozygous samples in the first-round amplification. Nevertheless, no expansion was evident.

A total of 93 samples of SUID cases were analyzed. No expansion was detected from either PCR amplification. The frequencies of 15 and 13 contractions were, respectively, 0.016 and 0.011, which were not significantly different from those in the control group. Moreover, the polyalanine tract comprises four synonymous codons. A polymorphism of synonymous codon was evident for both contraction alleles.

## Sequencing of *PHOX2B* for SUID subjects

For the SUID group, the whole gene were extensively sequenced. Several substitutions were detected in the entire gene (Fig 1). As a characteristic substitution, c.905A>C was detected in heterozygote, which was expected to be a non-synonymous substitution of p.Asn302Thr (Fig 2). The substitution has been registered with an accession number of rs779068107 (NM_003924.4) with an annotation of 'clinical significance unknown', for which a comment is accompanied by a potential effect to pathogenesis because of the highly conserved residue of asparagine. Furthermore, the APLP method was developed to detect the substitution of rs779068107 (Fig 3), but any other substitution was detected among the SUID and control groups.

In addition, for SNP (rs28647582) in intron 2, of which detection was performed using the APLP method (Fig 3), no significant allelic difference was obtained in the two groups of SUID and the control (Table 2).

## Case

This carrier of rs779068107, a 32-day-old male infant, was found unexpectedly to have no response in bed 3 h after feeding at midnight. At a hospital 30 min later, his death was confirmed. The baby had been delivered at 38 weeks. He weighed 3130 g with Apgar scores of 9 at 1 min and 10 at 5min. No record indicated an apneic event after birth. Potential hearing impairment was found in a screening, but no clinical examination had been done. His parents, both in their twenties, and a 2-year-old sister were healthy. Negative findings were obtained at autopsy, except for general congestion. Furthermore, no critical nucleotide substitutions for cardiovascular diseases, such as *KCNQ1*, *KCNH2* and *SCN5A*, were found in the comprehensive NGS analysis (supplementary data). SIDS was suspected as the cause of death after the initial examinations. In addition, the polyalanine repeat genotype was 20-times homozygous for the analyses described herein.

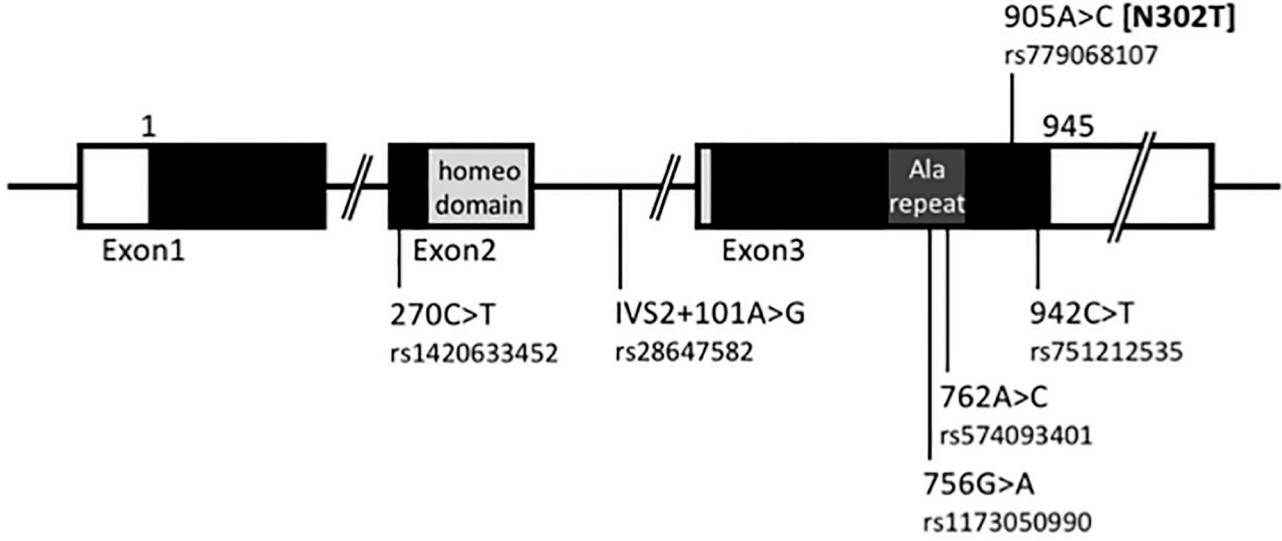

**Fig 1. Scheme of detected nucleotide substitutions in *PHOX2B* in SUID subjects (*n* = 93).**

## Discussion

To arrive at a diagnosis of SIDS, several disorders must be excluded. Molecular analysis has given forensic pathologists opportunities to find out potential physiological disorders of arrhythmic diseases such as long QT syndrome [20]. This approach is now known as molecular autopsy [21]. Actually, CCHS is a genetically detectable disease that presents difficulties for obtaining definitive morphological evidence in routine postmortem examinations [22].

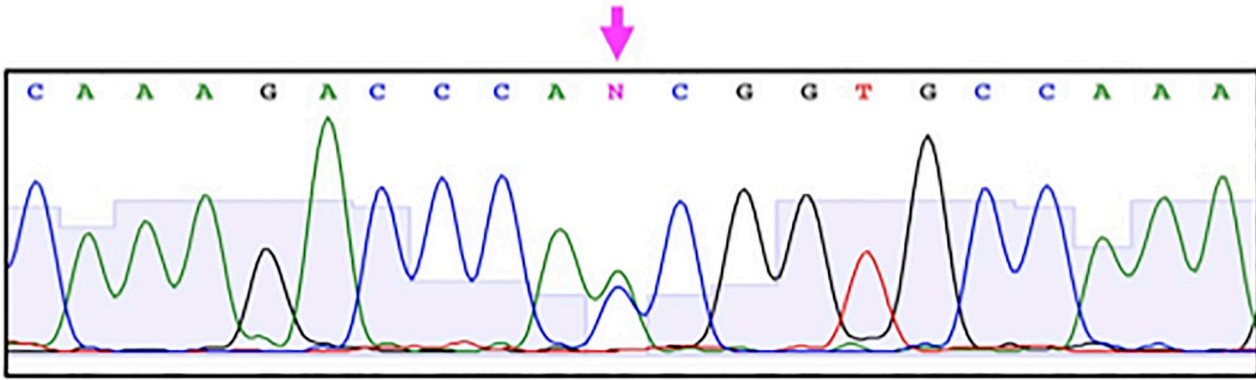

**Fig 2. Electropherogram of heterozygous 905A>C in *PHOX2B*.**

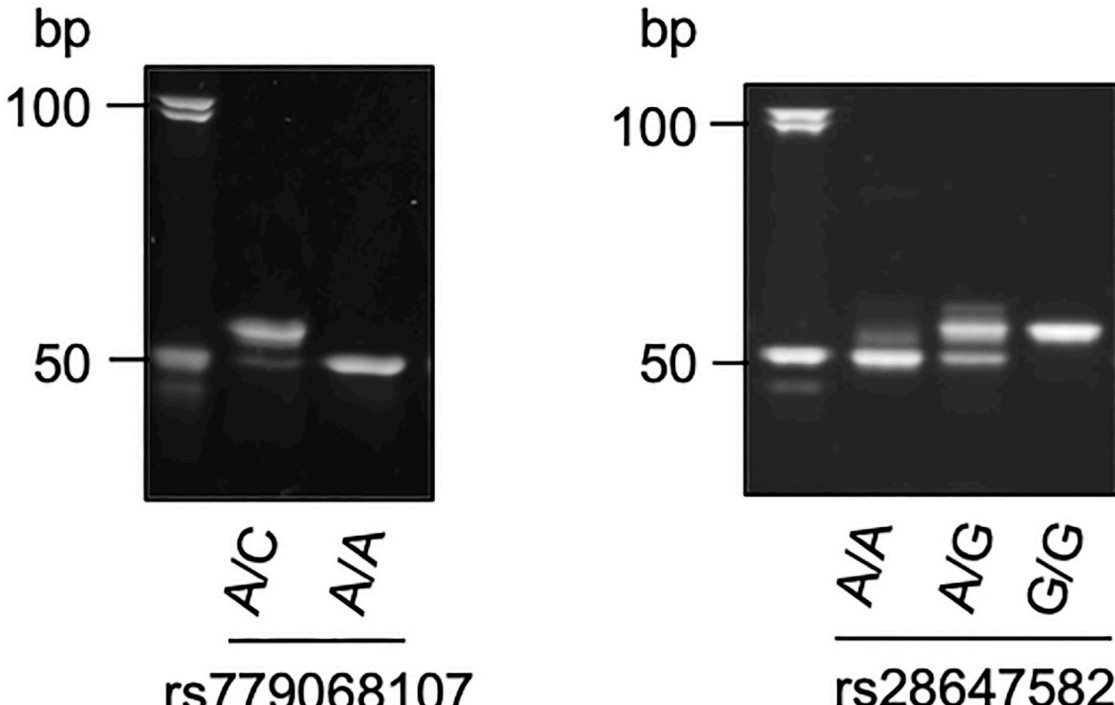

**Fig 3. APLP method to detect two substitutions of rs779068107 (left) and rs28647582 (right).** Amplified products were electrophoresed in 12% polyacrylamide gel, followed by ethidium bromide staining. Molecular size markers are in the left lane.

For this study, analyses were performed on a large scale to examine the involvement of repeat expansions and NPARMs in autopsy cases of SUID. No expansion was found in both SUID and control groups. An earlier publication showed positive correlation of repeat contraction with unclassified infant death [10]. By contrast, the present study revealed no significant relation, yielding similar results to that reported by Poetsch et al. [13]. The other report judged the 13-times repeat to be pathogenic [11], but this allele is distributed in healthy individuals even with low frequency, as found in this study. Clinical observations showed that contractions are not pathogenic [2, 7, 8]. The case control study by Bachetti et al. also showed no correlation of SIDS/SUID to the repeat contractions [12]. Therefore, it is reasonable to conclude that contractions are not associated with occurrence of SUID.

SIDS/SUID risk is still likely defined by multifactorial genetic and environmental interactions [23, 24]. In one of such subjects, we detected the missense mutation in *PHOX2B*. Symptoms such as apnea and cyanosis are typically noticed in the first 48 hr of life, but more patients with NPARMs are known to be evident outside of the newborn period, known as late-onset CCHS [25]. Moreover, as life-threatening events, bradycardia and sinus pauses potentially occur in addition to apnea during sleep [26–28]. Zhou et al. recently demonstrate that

**Table 2. Frequency (number) of genotypes and alleles of SNP (rs28647582) in intron 2 in the SUID and control groups.**

|  | Genotype | | | Allele | |
|---|---|---|---|---|---|
|  | A/A | A/G | G/G | A | G |
| SUID (n = 93) | 0.667 (62) | 0.280 (26) | 0.054 (5) | 0.806 (150) | 0.194 (36) P = .92 |
| Control (n = 942) | 0.653 (615) | 0.314 (296) | 0.033 (31) | 0.810 (1526) | 0.190 (358) |

phenotypic manifestations of NPARMs are associated with their variant type, location, and effect on transcripts [29]. Although the subject described for the present case seemed to have no episode of respiratory disorder after birth, the degree of severity in the respiratory and cardiovascular defect varies among cases [30, 31]. We infer that SUID cases might have causal NPARMs, rather than expansions.

The majority of CCHS-associated NPARMs are found at the ends of exons 2 and 3 [4]. The rs779068107 variant is also located near the C-terminal of molecules, and this should be the first clinical case. Though it is registered with annotation like 'clinical significance unknown', we think that this substitution might be phenotypically pathogenic because of the replacement of highly conserved amino acid residue,

In general, the impaired transfer of abnormal longer protein molecules through the nuclear membrane results in an excess in the cytoplasm to form intracellular aggregates. However, the molecular mechanism of NPARM is distinct from the intracellular localization. Phenotypes of the patients carrying NPARM are variable, and they might depend on mutant protein effects [3, 32]. For instance, a dominant negative effect might be derived from direct interference of the mutated protein with the wild-type activity. Gain-of-function effects might appear by altered expressional control in the original intranuclear distribution [2]. The substituted threonine residue potentially serves as a substrate for phosphorylation and glycosylation, which might exhibit such harmful effects.

Regarding SNP (rs28647582) in intron 2, significant associations with Hirschsprung disease, neuroblastoma and Wilms tumor have been demonstrated in a couple of studies [33, 34]. However, no association with SUID occurrence was obtained in this study. Liang et al. also showed no association of rs28647582 with Hirschsprung disease [35]. This SNP was not linked to polyalanine repeat number and other substitution in this conservative gene. Therefore, we think that this SNP does not have any sufficient effects to function and translated volume of *PHOX2B*.

In conclusion, our attempts highlight the importance of extensive search for the whole gene including non-polyalanine repeat regions. To date, evidence has indicated arrhythmia diseases such as QT elongation, and metabolic diseases such as fatty acids β-oxidation deficiency as prevalent in forensic autopsy cases of SIDS/SUID. Results of this study indicate the additional possibility of CCHS as a causal disorder for SIDS/SUID.

## Supporting information

**S1 Data.**
(XLS)

## Acknowledgments

We thank Dr. Takuma Yamamoto, Department of Legal Medicine, Hyogo College of Medicine, for his helpful discussion.

## Author Contributions

**Conceptualization:** Motoki Osawa.

**Data curation:** Eriko Ochiai.

**Investigation:** Atsushi Ueda, Haruaki Naito.

**Methodology:** Atsushi Ueda, Motoki Osawa.

**Project administration:** Motoki Osawa.

**Resources:** Motoki Osawa, Eriko Ochiai, Yu Kakimoto.

**Writing – original draft:** Atsushi Ueda.

**Writing – review & editing:** Motoki Osawa.

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
