## [Decision Letter · Decision Letter 0]

15 Feb 2022

PONE-D-22-00562Non-polyalanine repeat mutation in PHOX2B is detected in autopsy cases of sudden unexpected infant deathPLOS ONE

Dear Dr. Osawa,

Thank you for submitting your manuscript to PLOS ONE. After careful consideration, we feel that it has merit but does not fully meet PLOS ONE’s publication criteria as it currently stands. Therefore, we invite you to submit a revised version of the manuscript that addresses the points raised during the review process.

Your paper was reviewed by an expert in the field and myself. Though the topic is interesting, several points need to be clarified. Please read the comments and address the issues accordingly. In addition, please clarify the following several points:

1. Line 144-145. Please clarify if the nomenclatures are correct (seems A>C is correct according to Fig 2)(https://www.ncbi.nlm.nih.gov/clinvar?term=((807706[AlleleID])OR(520167[AlleleID]))).

2. Line 146-148. "As clinical significance at the site, it was annotated to that observed in a patient with CCHS-Hirschsprung disease (Haddad syndrome)."

Please place citation(s) for this sentence. If this information is from ClinVar (ID 535771 or 822948), please check the clinical status. "Affected status is unknown" in ClinVar means some individual(s) were tested for certain conditions including Haddad syndrome (e.g., https://www.ncbi.nlm.nih.gov/clinvar/variation/822948/evidence/). This is confusing, but actual clinical conditions (i.e., affected or not) are rarely stated in ClinVar. 

3. Line 149-151. Regarding rs28647582, the MAF (T>C) is very high in general populations (C=0.382239 in gnomAD). Thus, it is unlikely this SNP is causative. However, some researchers believe this can be a risk factor for SUD (https://www.ncbi.nlm.nih.gov/CBBresearch/Lu/Demo/LitVar/#!?query=rs28647582). This may be discussed (either agree or disagree). 4. Line 155-156. Please describe more detail regarding the findings of poly-alanine repeat. Is this condition associated with the phenotype of SIDS? If so, it would be better to add more data and/or discussion. 5.  Please add a statement regarding pathogenic variants of LQTS (LQT1-3) and Brugada syndrome (*SCN5A*) in the case presentation (page 10). I think excluding variants of congenital arrhythmia will make this study's finding more convincing. If data is unavailable, please state it as a limitation in Discussion section. 

We look forward to receiving your revised manuscript.

Kind regards,

Tomohiko Ai, M.D., Ph.D.

Academic Editor

PLOS ONE

Journal Requirements:

Reviewers' comments:

Reviewer's Responses to Questions

**Comments to the Author**

1. Is the manuscript technically sound, and do the data support the conclusions?

Reviewer #1: Yes

2. Has the statistical analysis been performed appropriately and rigorously? 

Reviewer #1: Yes

3. Have the authors made all data underlying the findings in their manuscript fully available?

Reviewer #1: Yes

4. Is the manuscript presented in an intelligible fashion and written in standard English?

Reviewer #1: Yes

5. Review Comments to the Author

Reviewer #1: The authors analyzed the entire nucleotide sequence of the phox2b locus in Japanese autopsy cases of SIDS/SUID and reported that cchs may be involved in the cause of death in one case with a nonsynonymous substitution. This is a valuable report on the involvement of cchs in sudden infant death syndrome.

Some comments are provided below.

It is desirable to present representative data of the APLP electrophoresis diagram for typing of rs779068107.

The minor allele frequency of rs779068107 in Japanese is required to be described. Compared to the minor allele frequency, do you consider the number of healthy controls to be adequate?

As stated in the manuscript, rs779068107 has been found in patients with Haddad syndrome, but the significance of this mutation in CLINVAR is UNCERTAIN SIGNIFICANCE. As stated in the manuscript, rs779068107 is found in patients with Haddad syndrome, but the significance of this mutation in CLINVAR is UNCERTAIN SIGNIFICANCE. Please discuss this, comparing the mutations around rs779068107 in the phox2b gene may help to solve this problem.

6. PLOS authors have the option to publish the peer review history of their article (what does this mean?). If published, this will include your full peer review and any attached files.

Reviewer #1: No

---

## [Author Response · Author response to Decision Letter 0]

30 Mar 2022

To the Reviewers' comments

Thank you very much for the detailed review to our submission. The raised comments were helpful to revise the manuscript. The reply is described below. Alterations are indicated in the revised text with track changes in red.

1. Line 144-145. Please clarify if the nomenclatures are correct (seems A>C is correct according to Fig 2) (https://www.ncbi.nlm.nih.gov/clinvar?term=((807706[AlleleID])OR(520167[AlleleID]))).

Reply

Both c.905A>G and c.905A>C are registered in the NCBI web site. As shown in Fig. 2, c.905A>C is correct. However, the authors made a mistake of the amino acid substitution of Asn302Ser in the original manuscript. We have corrected it to Asn302Thr in line 160 of page 11.

2. Line 146-148. "As clinical significance at the site, it was annotated to that observed in a patient with CCHS-Hirschsprung disease (Haddad syndrome)."

Please place citation(s) for this sentence. If this information is from ClinVar (ID 535771 or 822948), please check the clinical status. "Affected status is unknown" in ClinVar means some individual(s) were tested for certain conditions including Haddad syndrome (e.g., https://www.ncbi.nlm.nih.gov/clinvar/variation/822948/evidence/). This is confusing, but actual clinical conditions (i.e., affected or not) are rarely stated in ClinVar.

Reply

As the reviewer pointed out, the web site says "Affected status is unknown" for the missense substitution. The careless description has been rewritten extensively in the revised version, in which the term of Haddad syndrome has been removed thoroughly. Please see lines 31 to 33 in page 2, lines 161 to 163 in page 11, and lines 219 to 220 in page 14. We have extensively rewritten it to the revised version as uncertain significance. Please see lines 33 to 35 in page 2, lines 163 to 165 in page 11, and lines 231 to 234 in page 15.

3. Line 149-151. Regarding rs28647582, the MAF (T>C) is very high in general populations (C=0.382239 in gnomAD). Thus, it is unlikely this SNP is causative. However, some researchers believe this can be a risk factor for SUD (https://www.ncbi.nlm.nih.gov/CBBresearch/Lu/Demo/LitVar/#!?query=rs28647582). This may be discussed (either agree or disagree).

Reply

The authors do not think that the substitution of rs28647582 is related to sudden unexpected infant death. The statement is added into the sections of abstract and discussion of the revised manuscript. Please see lines 28 to 29 in page 2, and lines 245 to 251 in page 16. 

In addition, the result of genotyping was separately described as Table 2 with division from the original Table 1. Please see lines 145 to 148 of pages 9 and 10, and lines 182 to 183 of page 12. 

As the reviewer found out in the web site, there are a couple of reports that showed positive and negative relations of rs28647582 SNP to Hirschsprung disease, neuroblastoma and Wilms tumor. We also added a new paragraph about these points into the revised text as well. Please see lines 245 to 251 in page 16, and nos. 33-35 of reference list of pages 22 to 23.

4. Line 155-156. Please describe more detail regarding the findings of poly-alanine repeat. Is this condition associated with the phenotype of SIDS? If so, it would be better to add more data and/or discussion.

Reply

According to the advice, the statement regarding poly-alanine repeat to SUID has been added into Discussion of the revised text more clearly. Please see line 208, and lines 214 to 215 in page 14.

5. Please add a statement regarding pathogenic variants of LQTS (LQT1-3) and Brugada syndrome (SCN5A) in the case presentation (page 10). I think excluding variants of congenital arrhythmia will make this study's finding more convincing. If data is unavailable, please state it as a limitation in Discussion section.

Reply

According to the advice from the reviewer, the authors performed the additional genetic testing using next generation sequencing once again. This revealed no critical exchanges in genes associated with the cardiovascular diseases, such as KCNQ1, KCNH2 and SCN5A, in spite of a number of synonymous and missense mutation including common SNPs. The procedure and result were added into the revise version. Please see lines 116 to 126 in page 8, and lines 192 to 195 in page 13.

Further, the raw data is newly added as ‘supplementary data’ with the predictive in silico analysis.

 

To Reviewer #1

Thank you very much for the detailed review to our submission. The raised comments were helpful to revise the manuscript. The reply is described below. Alterations are indicated in the revised text with track changes in red.

1. It is desirable to present representative data of the APLP electrophoresis diagram for typing of rs779068107.

Reply

According to the advice from the reviewer, the amplified product length polymorphism (APLP) electrophoresis diagram for rs779068107 typing was added into the revised manuscript as Fig. 3. The authors showed the diagram for rs28647582 as well. Please see lines 165 to 167, and lines 174 to 176 in pages 11 to 12, and lines 178 to 179 in page 12.

2. The minor allele frequency of rs779068107 in Japanese is required to be described. Compared to the minor allele frequency, do you consider the number of healthy controls to be adequate?

Reply

This c.905A>C had not been found in the Japanese population as long as referring to the public genome bank. The authors have deposited this substitution to the DDBJ genome bank. Therefore, it was difficult to calculate the frequency in the population. The authors added this point into the revised version. Please see lines 231 to 234 in page 15.

3. As stated in the manuscript, rs779068107 has been found in patients with Haddad syndrome, but the significance of this mutation in CLINVAR is UNCERTAIN SIGNIFICANCE. Please discuss this, comparing the mutations around rs779068107 in the phox2b gene may help to solve this problem.

Reply

As the reviewer pointed out, the substitution was seemingly not confirmed in a clinical patient of Haddad syndrome as long as seeing the web site. It was our fault. The careless description has been rewritten in the revised version, in which the term of Haddad syndrome has been removed thoroughly. Please see lines 31 to 33 in page 2, lines 161 to 163 in page 11, and lines 219 to 220 in page 14. We have extensively rewritten it to the revised version as uncertain significance. Please see lines 33 to 35 in page 2, lines 163 to 165 in page 11, and lines 231 to 234 in page 15.

---

## [Decision Letter · Decision Letter 1]

18 Apr 2022

Non-polyalanine repeat mutation in PHOX2B is detected in autopsy cases of sudden unexpected infant death

PONE-D-22-00562R1

Dear Dr. Osawa,

We’re pleased to inform you that your manuscript has been judged scientifically suitable for publication and will be formally accepted for publication once it meets all outstanding technical requirements.

Kind regards,

Tomohiko Ai, M.D., Ph.D.

Academic Editor

PLOS ONE

Additional Editor Comments (optional):

Reviewers' comments:

Reviewer's Responses to Questions

**Comments to the Author**

1. If the authors have adequately addressed your comments raised in a previous round of review and you feel that this manuscript is now acceptable for publication, you may indicate that here to bypass the “Comments to the Author” section, enter your conflict of interest statement in the “Confidential to Editor” section, and submit your "Accept" recommendation.

Reviewer #1: All comments have been addressed

2. Is the manuscript technically sound, and do the data support the conclusions?

Reviewer #1: Yes

3. Has the statistical analysis been performed appropriately and rigorously? 

Reviewer #1: Yes

4. Have the authors made all data underlying the findings in their manuscript fully available?

Reviewer #1: Yes

5. Is the manuscript presented in an intelligible fashion and written in standard English?

Reviewer #1: Yes

6. Review Comments to the Author

Reviewer #1: The manuscript reports on non-polyalanine repeat mutation in PHOX2B in autopsy cases of sudden unexpected infant death. The manuscript has been revised well. The reviewer thinks this manuscript will be acceptable.

7. PLOS authors have the option to publish the peer review history of their article (what does this mean?). If published, this will include your full peer review and any attached files.

Reviewer #1: No

---

## [Editor Report · Acceptance letter]

21 Apr 2022

PONE-D-22-00562R1 

Non-polyalanine repeat mutation in *PHOX2B* is detected in autopsy cases of sudden unexpected infant death 

Dear Dr. Osawa:

I'm pleased to inform you that your manuscript has been deemed suitable for publication in PLOS ONE. Congratulations! Your manuscript is now with our production department. 

Kind regards, 

on behalf of

Dr. Tomohiko Ai 

Academic Editor

PLOS ONE